# Microstructure Evolution of the Near-Surface Deformed Layer and Corrosion Behavior of Hot Rolled AA7050 Aluminum Alloy

**DOI:** 10.3390/ma16134632

**Published:** 2023-06-27

**Authors:** Ergen Liu, Qinglin Pan, Bing Liu, Ji Ye, Weiyi Wang

**Affiliations:** 1School of Materials Science and Engineering, Central South University, Changsha 410083, China; ergenliu@163.com (E.L.); weiyiwang@163.com (W.W.); 2Light Alloy Research Institute, Central South University, Changsha 410083, China

**Keywords:** AA7050 aluminum alloy, hot rolling, near-surface deformed layer, microstructure evlution, corrosion behavior

## Abstract

The current study investigates the influence of hot rolling on the microstructure evolution of the near-surface region on AA7050 aluminum alloy and the corrosion performance of the alloy. It is revealed that hot rolling resulted in grain refinement in the near-surface region, caused by dynamic recrystallization, and equiaxed grains less than 500 nm can be clearly observed. Fibrous grains were evident in the hot rolled AA7050 aluminum alloy with relatively lower rolling temperature or larger rolling reduction, caused by the more severe elemental segregation at grain boundaries, which inhibited the progression of dynamic recrystallization. The density of the precipitates in the fibrous grain layer was higher, compared with those in the equiaxed grain layer, due to the increased dislocation density, combined with more severe elemental segregation, which significantly promoted the nucleation of precipitates. With the co-influence exerted by low density of precipitates and dislocations on the improvement of the corrosion performance of the alloy, the rolled AA7050 alloy with decreased density of precipitates and dislocations exhibited better corrosion resistance.

## 1. Introduction

Aluminum alloy is one of the most widely used nonferrous metal structural materials in industry. Due to its high power-to-weight ratio, low cost, high wear resistance and other good properties, aluminum alloy has been widely used in many structural parts of aviation, aerospace, automobile, machinery manufacturing, and shipbuilding [1,2,3,4,5]. Rolling is often employed during manufacture of such alloys. Thus, the microstructural evolution caused by rolling should be considered, as it will influence the properties and finally limit the better applications of such alloys. Guo et al., [6] found that the fraction of the low-angle grain boundaries (LAGBs) increased and S phase precipitates located at the grain boundaries were discontinuous when Al-Cu-Mg-Ag alloy were rolled at a lower temperature. Kumar et al., [7] proposed that dislocation cells, un-recrystallized grains and nano-sized precipitates were evident on cold rolled AA6082 aluminum alloy. Wang et al., [8] reported the presence of shear bands on cold rolled AA7050 aluminum alloy, and such shear bands became wide with the increase of the rolling reduction. Liu et al., [9] reported that the textures obviously changed in Al-Cu alloy due to the increase of the cold rolling reduction. Based on the previous study, extensive work has been done regarding the effect of rolling on the microstructure of aluminum alloys [6,7,8,9,10,11,12]. However, the near-surface region has rarely been discussed, which is directly in contact with the service environment and will determine the properties of rolled aluminum alloys to a large extent.

In recent years, it has been found that severe plastic deformation, including machining [13], orthogonal cutting [14], laser shock processing [15] and other processes significantly influence the near-surface microstructure of aluminum alloy, leading to the heavily deformed layers characterized by ultra-fine grains. Within such deformed layers, obvious residual stresses, high density of dislocations and element redistribution have also been observed which, in combination, affect the corrosion performance of aluminum alloys [16,17,18,19,20,21]. The near-surface deformed layer on the weld joint of Al-Cu-Li alloy is more active than the bulk alloy, and anodic dissolution usually occurs preferentially in such regions of the weld joint subjected to corrosive environment [22]. Liu et al., [23] found that element segregation in the near-surface deformed layer of AA7075-T6 aluminum alloy promoted the localized corrosion along the grain boundaries. Saklakoglu et al., [15] reported that high level of work-hardening and compressive residual stress were observed in the near-surface deformed layer on AA6061-T6 aluminum alloy treated by laser shock peening, which improved the resistance against pitting corrosion. Based on the previous study, the near-surface deformed layers have both advantages and disadvantages on the corrosion performance of aluminum alloys, depending on the microstructure formed during the severe deformation. Since hot rolling is normally employed during fabrication of aluminum alloys and corrosion which starts the surface of the alloy is always one of the main concerns of such alloys during service, it is necessary to investigate the corrosion behavior of hot rolled 7xxx series aluminum alloys, considering the impact of near-surface deformed layer.

The main purpose of the current work is to investigate the microstructure evolution of the near-surface deformed layer and the corrosion performance of the hot rolled AA7050 aluminum alloy, and to establish the correlation between them, which contributes to improving the performance and service life of hot rolled AA7050 aluminum alloy as a structural material.

## 2. Materials and Methods

### 2.1. Materials

AA7050 aluminum alloy ingot was cut into alloy samples with dimensions of 200 × 110 × 60 mm. The chemical composition of the alloy is shown in Table 1. After homogenization annealing at 465 °C for 24 h, the alloy ingots were hot rolled at the temperature of 380 °C (1#), 420 °C (3#), 450 °C (5#) with rolling reduction of 66.7%, and at the temperature of 420 °C with the rolling reductions of 83.3% (2#), 66.7% (3#), 50% (4#), respectively. Hot rolling of the alloy was carried out using the L6500 two rolling mill, of which the roll diameter and speed were 420 mm and 0.36 m/s, respectively. The reduction of each rolling pass was less than 5 mm. The rolling parameters of each sample are summarized in Table 2.

### 2.2. Mircrostructure Characterization

Transmission electron microscopy (TEM) samples were prepared by focused ion beam (FIB), using a FEI Helios NanoLab G3 UC dual-beam focused ion beam and a scanning electron microscope. The voltage and current during sample preparation were 5~30 kV and 41 pA~10 nA, respectively. A Tecnai G^2^ 20 transmission electron microscope and a Talos F200X field emission transmission electron microscope equipped with the EDS detector were employed to observe the microstructure of the near-surface deformed layers on the alloy samples subjected to hot rolling under different conditions. The morphology of the corroded surface was observed by a SIRION200 field emission scanning electron microscope equipped with the EDS detector.

### 2.3. Corrosion Test

Potentiodynamic polarization, open circuit potential (OCP) and electrochemical impedance spectroscopy (EIS) measurements were conducted in 3.5 wt.% NaCl solution on a Multi Autolab M204 electrochemical workstation. The alloy samples served as the working electrode, while the reference electrode (RE) and counter electrode (CE) were saturated calomel electrode (SCE) and platinum wire, respectively, in the three-electrode system. The testing area involved in the electrochemical measurements was 10 × 10 mm^2^. The duration of the immersion tests were 6 h, 12 h, 18 h, and 24 h, respectively. The OCP test was carried out for 3600 s. The scan range selected for potentiodynamic polarization test was from −1.1 V (SCE) to −0.3 V (SCE), with a scan speed of 1 mV/s. The EIS measurement was carried out from 10^5^ Hz to 0.01 Hz and the sinusoidal voltage amplitude was 10 mV.

## 3. Results

### 3.1. Microstructure of Near-Surface Deformed Layer on Hot Rolled AA7050 Aluminum Alloy

Figure 1 displays the High Angle Annular Dark Field (HAADF) micrographs of the near-surface region on AA7050 aluminum alloy subjected to hot rolling under different conditions. Figure 1a,b are the near-surface regions of 1# and 2# samples, respectively, indicating that the near-surface deformed layers on the two alloy samples were mainly composed of an equiaxed grain layer, which was near the rolled surface, and a fibrous grain layer beneath the equiaxed grain layer. The thickness of the equiaxed grain layer was approx. 3.5 µm and 2.5 µm, and the dimension of the equiaxed grains was approx. 400 nm and 500 nm, respectively, on 1# and 2# samples. The boundaries between the equiaxed grain layers and the fibrous grain layers are indicated by curved lines in Figure 1a,b. The density of precipitates in the fibrous grain layer was significantly higher than that in the equiaxed grain layer. Figure 1c–e are the near-surface deformed layers of 3#, 4# and 5# samples, respectively. It is revealed that the near-surface deformed layers were mainly characterized by equiaxed grains, with the dimensions of approx. 400 nm, 500 nm, and 500 nm, respectively. The thickness of the near-surface deformed layers was more than 5.5 µm, and the precipitates can be observed uniformly distributed in the near-surface deformed layers on such samples.

Figure 2 shows the HAADF micrographs, showing distribution of precipitates in the top regions of the near-surface deformed layers on the rolled AA7050 aluminum alloy under different conditions. It can be observed that such precipitates vary from 20–200 nm distributed at the equiaxed grain boundaries. It is also evident that the equiaxed grain boundaries were bright in the HAADF micrographs. Due to the high dependence of the brightness of the HAADF micrographs on atomic number, its brightness is the relative abundance of heavy alloy elements, suggesting the segregation of heavier elements such as Cu and Zn compared with aluminum.

Figure 3 exhibits the HAADF micrographs, showing distribution of precipitates in the bottom regions of the near-surface deformed layers on the rolled AA7050 aluminum alloy under different conditions. The precipitates can clearly be observed in the fibrous grain layers, indicated in Figure 3a,b. In addition, the bright bands also appeared at the fibrous grain boundaries, suggesting the presence of segregation of alloy elements at such locations. In the bottom regions of the near-surface deformed layers on the 3#, 4# and 5# samples (Figure 3c–e), the precipitates can be observed in the equiaxed grain boundaries. The brightest bands at the grain boundaries disappeared, indicating that the alloy elements segregation was not obvious at the grain boundaries on the rolled alloy samples under such conditions. From Figure 2 and Figure 3, it can be observed that when the hot rolling parameters change, element segregation also changes, forming the near-surface deformed layer near the rolling surface to the entire near-surface deformed layer.

In order to further determine the composition of the elemental segregation, the 1# sample (380 °C, 66.7%) was selected for EDS mapping analysis, as shown in Figure 4. It is evident that the grain boundaries were rich in Cu, Zn and Mg in the near-surface deformed layer on the 1# sample, which also suggests that the bright bands were caused by the accumulation of Mg, Zn and Cu atoms. In addition, the precipitates observed in the near-surface deformed layer on 1# sample mainly contained Cu, Mg and Zn elements, as indicated in Figure 4, which might be the η (MgZn_2_) phase enriched with Cu. The O element was mainly concentrated on the rolling surface, which was caused by the formation of oxides and promoted by the high temperature during hot rolling [24].

In order to further determine whether the precipitated phase was a η phase, we conducted a more in-depth study on it. Figure 5a displays the lattice image of the precipitates observed in the near-surface deformed layers along the [1$\overset{–}{1}$2]_Al_ axis. The lattice parameters of the precipitate phase were measured as follows: a = 0.472 nm, c = 0.893 nm, respectively. Meanwhile, the lattice parameters of the η phase are as follows: a = 0.504 nm, c = 0.828 nm [25]. Figure 5b shows the enlarged image in Figure 5a. It can be found from the HAADF micrograph that the stacking structure was mainly R/R^−1^, with occasional presence of an R^−1^R^−1^ stacking layer-structure. Previous studies pointed out that the R/R^−1^ stacking layer-structure often existed in η phase [26,27,28,29], and even an R/R^−1^ stacking layer-structure is presumed to exist in all types of the η phase [26]. Chung et al., [29] believed that the formation of the R^−1^R^−1^ stacking structure was typical in the η phase. Therefore, based on the experimental results, it can be determined that the precipitated phase is the η phase.

In addition to precipitation of larger precipitates at grain boundaries (Figure 1, Figure 2 and Figure 3), the fine precipitates are also precipitated inside the grains. Figure 6 displays the TEM micrographs showing the top regions of the near-surface layers, which are near the rolling surfaces on AA7050 aluminum alloy subjected to hot rolling under different conditions. It can be revealed that precipitate-free zone was not obviously adjacent to the grain boundaries in the near-surface deformed layers on the hot rolled AA7050 aluminum alloy under various conditions. At the same rolling temperature of 420 °C (2#, 3#, and 4# samples), the density of precipitates did not vary obviously with the increase of rolling reduction (Figure 6b–d). Meanwhile, the grain boundary precipitates of the three samples were discontinuous, and fine precipitates with dimensions of approx. 5–10 nm can be clearly observed. With the same rolling reduction of 66.7% (1#, 3#, and 5# samples), with the increase of rolling temperature, the density of such fine precipitates decreased in the order of 5# (450 °C) > 1# (380 °C) > 3# (420 °C), and the size of such fine precipitates also decreased, as indicated in Figure 6a,c,e. The grain boundary precipitates were distributed continuously at the lower rolling temperature (380 °C) and discontinuously at the higher rolling temperature (450 °C).

Figure 7 displays the TEM micrographs showing the top regions of the near-surface layers near the rolling surfaces on AA7050 aluminum alloy subjected to hot rolling under different conditions. It can be observed that dynamic recrystallization occurred, and the dislocations were annihilated and reordered by slipping and climbing in all samples, resulting in the disappearance of dislocations and the formation of dislocation cells in the structure. Compared with 3# sample (66.7%), the density of dislocations was reserved after dynamic recrystallization was higher in 2# (83.3%) and 4# samples (50%), as indicated by the arrows in Figure 7b,d. However, there was no significant difference in the dislocation density of sample 1#, 3# and 5# (Figure 7a,c,e).

### 3.2. Electrochemical Corrosion Behavior of Hot Rolled AA7050 Aluminum Alloy

The OCP curve of hot rolled AA7050 aluminum alloy in 3.5 wt.% NaCl solution are displayed in Figure 8. According to Figure 8a, with the same rolling reduction of 66.7%, the OCP of 1# (380 °C), 3# (420 °C) and 5# (450 °C) samples varied obviously, with the values of −815.140 mV (SCE), −802.343 mV (SCE) and −844.650 mV (SCE), respectively, among which the OCP of 3# sample was the most positive, and the OCP of 5# sample was the most negative. The OCP values of such samples are listed in Table 3. It is clearly revealed that, with the same rolling reduction, the corrosion resistance of the aluminum alloy, rolled at different temperatures, was in the sequence: 3# > 1# > 5#. With the same hot rolling temperature of 420 °C, the OCP values of 2# (88.3%), 3# (66.7%) and 4# (50%) samples were −815.081 mV (SCE), −802.343 mV (SCE) and −804.718 mV (SCE), respectively, as evidenced in Figure 8b, indicating that the corrosion resistance of the hot rolled aluminum alloy was in the order of 3# > 4# > 2#.

Figure 9 shows potentiodynamic polarization curves of hot rolled AA7050 aluminum alloy immersed in 3.5 wt.% NaCl solution. The corrosion potential and corrosion current density were calculated based on the results of the measurements, which are shown in Table 3. Figure 9a exhibits that with the same rolling reduction of 66.7%, the corrosion current density (I_corr_) of the rolled alloy samples gradually decreased in the order of 5# (450 °C) > 1# (380 °C) > 3# (420 °C), and the corrosion potential gradually decreased in the order of 3# > 1# > 5#, suggesting that the corrosion resistance of the hot rolled aluminum alloy was in the order of 3# > 1# > 5#. Additionally, at the same hot rolling temperature of 420 °C, corrosion current density gradually decreased in the order of 2# (83.3%) > 4# (50%) > 3# (66.7%), and the corrosion potential gradually decreased in the order of 3# > 4# > 2#, as shown in Figure 9b, indicating that the corrosion resistance of the hot rolled aluminum alloy was in the order of 3# > 4# > 2#.

Figure 10 shows the Nyquist diagrams of AA7050 aluminum alloy immersed in 3.5 wt.% NaCl solution. It is clearly revealed that the capacitive arc was evident at high frequency in the Nyquist diagram of the 1# sample, the inductive arc can be observed at the low frequency, and only the capacitive arc was obvious at high frequency in the Nyquist diagrams of the other four samples. The radius of the capacitive arc of the alloy samples rolled with the same reduction of 66.7% followed the order of 3# (420 °C) > 1# (380 °C) > 5# (450 °C), and that of the alloy samples rolled at the same temperature of 420 °C followed the order of 3# (66.7%) > 4# (50%) > 2# (83.3%), as evidenced in Figure 10a,b, respectively. Generally, the larger the radius of the capacitive arc is related to the better corrosion resistance of the alloy. Therefore, according to the experimental results, it is clearly evident that the 3# sample (420 °C, 66.7%) was the most corrosion resistant among the five samples.

Figure 11 and Figure 12 show the corrosion morphology of rolled AA7050 aluminum alloy immersed in 3.5 wt.% NaCl solution for 6, 12, 18, and 24 h under different rolling conditions. It can be observed in Figure 11 and Figure 12 that in the early stage of the immersion test (6 h), corrosion was not obvious on the rolled alloy surface of all samples. However, when the immersion time was extended to 12–24 h, compared with the 3# sample, the density of corrosion pits and corrosion products of the other samples significantly increased while corrosion was extended to increased areas on such samples. Based on the results, the 3# samples exhibited the best corrosion resistance, which was also consistent with the results of the potentiodynamic polarization and the EIS experiment.

## 4. Discussion

### 4.1. Effect of Hot Rolling Parameters on the Near-Surface Deformed Layer

In summary, after hot rolling, the size of equiaxed grains in the near-surface regions on all the rolled samples did not exceed 500 nm, and the precipitates were uniformly distributed in the equiaxed grain layers. However, the density of precipitates in the fibrous grain layers was significantly higher than that in the equiaxed grain layers (1# and 2#). Additionally, variation in hot rolling conditions led to varying degrees of grain boundary segregation. The segregation in the near-surface regions on 3#, 4#, and 5# samples was mainly concentrated at the top of the near surface deformed layers, while in 1# and 2# samples, element segregation was distributed throughout the near-surface deformed layers.

The grain size of Al-Zn-Mg-Cu alloys treated by conventional thermo-mechanical processing is normally in the micron-scale [30]. Obviously, in the current study, under the co-influence of elevated temperature and high level of strain caused by severe deformation during hot rolling, dynamic recrystallization occurred in the alloy. Consequently, the deformation introduced dislocations with increased density which moved to form the sub-grain structures with relatively low-angle grain boundaries. Such sub-grains continuously rotated, and the large fibrous grains were eventually divided into fine equiaxed grains with dimensions less than 500 nm which were evident in the near-surface layers on the hot rolled AA7050 aluminum alloy under different conditions (Figure 1, Figure 2 and Figure 3).

During the hot rolling process, grains in AA7050 aluminum alloy were elongated along the rolling direction, leading to the formation of fibrous grains. Such fibrous grains in the near-surface deformed layers were mostly subdivided into equiaxed grains caused by dynamic recrystallization during hot deformation, as evidenced in samples 3#, 4#, and 5#. However, fibrous grains were still evident at the bottom regions of the near-surface deformed layers in samples 1# and 2#. Such phenomena can be explained by the elemental segregation at grain boundaries in the near-surface deformed layers on samples 1# and 2# (Figure 2 and Figure 3), and comparing that with the other samples. The solute atoms at grain boundaries significantly facilitated the occurrence of dynamic recovery and accelerated the annihilation and reordering of dislocations, thus reducing the deformation energy storage of aluminum alloys [30] which is the driving force of dynamic recrystallization. Consequently, dynamic recrystallization was not obvious in such regions, and fine equiaxed grains can hardly be observed.

Dynamic recrystallization introduced in equiaxed grain layers can also been observed adjacent to the fibrous grain layer in the 1# and 2# samples at the top of the near-surface regions near the rolling surfaces. Since the temperature and strain gradually decreased from the rolling surface to the bulk alloy, and the top region of the near-surface deformed layers experienced the highest level of temperatures and strains, the high density of dislocations led to the formation of sub-grains and finally the fibrous grains were divided into fine grains.

As evidenced in Figure 1, Figure 2 and Figure 3, the density of the precipitates was higher in the fibrous grain layers in the near-surface regions with the presence of both the equiaxed grain layer and the fibrous grain layer (1# and 2# samples). Such phenomena can be explained by the higher density of defects, such as vacancies and dislocations, promoting the diffusion of the solute atoms, providing ideal locations for the nucleation of the precipitates and the segregation of elements at the grain boundaries in the fibrous grain layer. Such dislocations largely promoted the nucleation of precipitates, and the segregation of elements also contributed to the formation of the precipitates rich in alloy elements. While in the equiaxed grain layer, most dislocations were consumed by dynamic recrystallization, resulting in fewer locations for nucleation of precipitates. Additionally, during the hot rolling process, extensive heat was generated as a result of the elevation of the temperature near the rolling surface, leading to the further dissolution of the precipitates. Consequently, this was relatively far from the rolling surface compared with the equiaxed grain layer, as evidenced in 1# and 2# samples.

### 4.2. Effect of Microstructure of Near-Surface Deformed Layer on Corrosion

It is well known that the corrosion of aluminum alloys starts from the surface region and is closely related to the surface condition. Microstructure evolution of the near-surface region, including the feature of the grains, the density, dimension and distribution of precipitates, as well as the density and distribution of dislocations directly influence the corrosion behavior of the alloy. Generally, corrosion preferentially occurs in the locations with high energy [31,32,33,34], such as the locations with high density of dislocations.

At the same rolling temperature of 420 °C, with the increase of rolling reduction, the density, dimension and distribution of precipitates in the near-surface region of the rolled samples showed no obvious variation (Figure 6), but the presence of dislocations retained after dynamic recrystallization led to increased stored energy, as shown in Figure 7. This significantly promoted the corrosion susceptibility of the rolled alloy. Thus, the corrosion resistance of 2# (83.3%) and 4# (50%) samples with higher dislocation density decreased compared with 3# (66.7%).

With the same rolling reduction of 66.7%, the density of such fine precipitates decreased in the order of 5# (450 °C) > 1# (380 °C) > 3# (420 °C). The density of dislocations was similar among the samples, as evidenced in Figure 6 and Figure 7. It is found that the corrosion potential of the Al matrix is −0.68 V (SCE), while that of the MgZn_2_ phase is approx. −1~−1.07 V (SCE) [30,35,36,37,38]. Consequently, the increase of the density of the precipitates results in the promotion of electrochemical inhomogeneity within the alloy, and the alloy was less corrosion resistant in the corrosion environment. At the same time, with the increase of the temperature, the distribution of grain boundary precipitates varied from continuous to discontinuous, indicating that the density of precipitates was an important factor affecting the corrosion behavior, compared with the distribution of precipitates.

## 5. Conclusions

In the current work, the hot rolling introduced near-surface deformed layer and its influence on corrosion behavior of the hot rolled AA7050 aluminum alloy was investigated. The main conclusions can be summarized as follows:Under the action of elevated temperature and strain introduced during hot rolling, dynamic recrystallization occurred in the near-surface region, resulting in the generation of equiaxed grains and refinement of grains.With the same rolling reduction of 66.7%, and with the reduction of rolling temperature or at the same rolling temperature of 420 °C, the increase of rolling reduction caused the segregation of Cu, Zn and Mg elements to become more serious. This occurred from the near-surface deformed layer near the rolling surface to the entire near-surface deformed layer.The density of precipitates in the fibrous grain layer was much higher than that in the equiaxed grain layer due to the elemental segregation and higher density of dislocations in such layers, which provided increased locations as well as higher concentration of solute atoms for nucleation of precipitates.The density of the precipitates and dislocations are the key factors affecting the corrosion properties of rolled alloys. With the same rolling reduction, the corrosion resistance mainly depends on the density of precipitates, since it significantly promotes the electrochemical inhomogeneity within the alloy. At the same rolling temperature, the corrosion resistance of the rolled AA7050 aluminum alloy is closely related to the density of dislocations, which also contributes to the initiation of corrosion.

## Figures and Tables

**Figure 1 materials-16-04632-f001:**
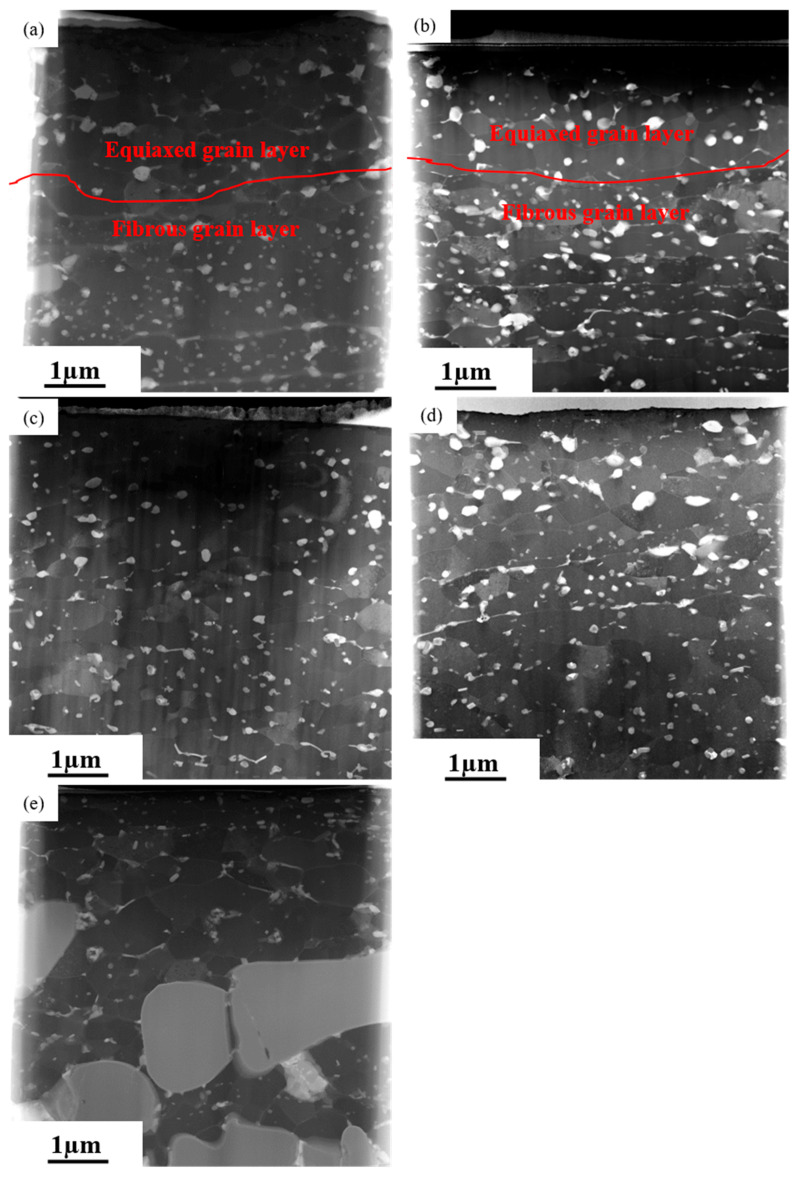
The HAADF micrographs of the near-surface deformed layers on AA7050 aluminum alloy subjected to hot rolling under different conditions: (**a**) 1# (380 °C, 66.7%); (**b**) 2# (420 °C, 83.3%); (**c**) 3 # (420 °C, 66.7%); (**d**) 4# (420 °C, 50%); (**e**) 5# (450 °C, 66.7%).

**Figure 2 materials-16-04632-f002:**
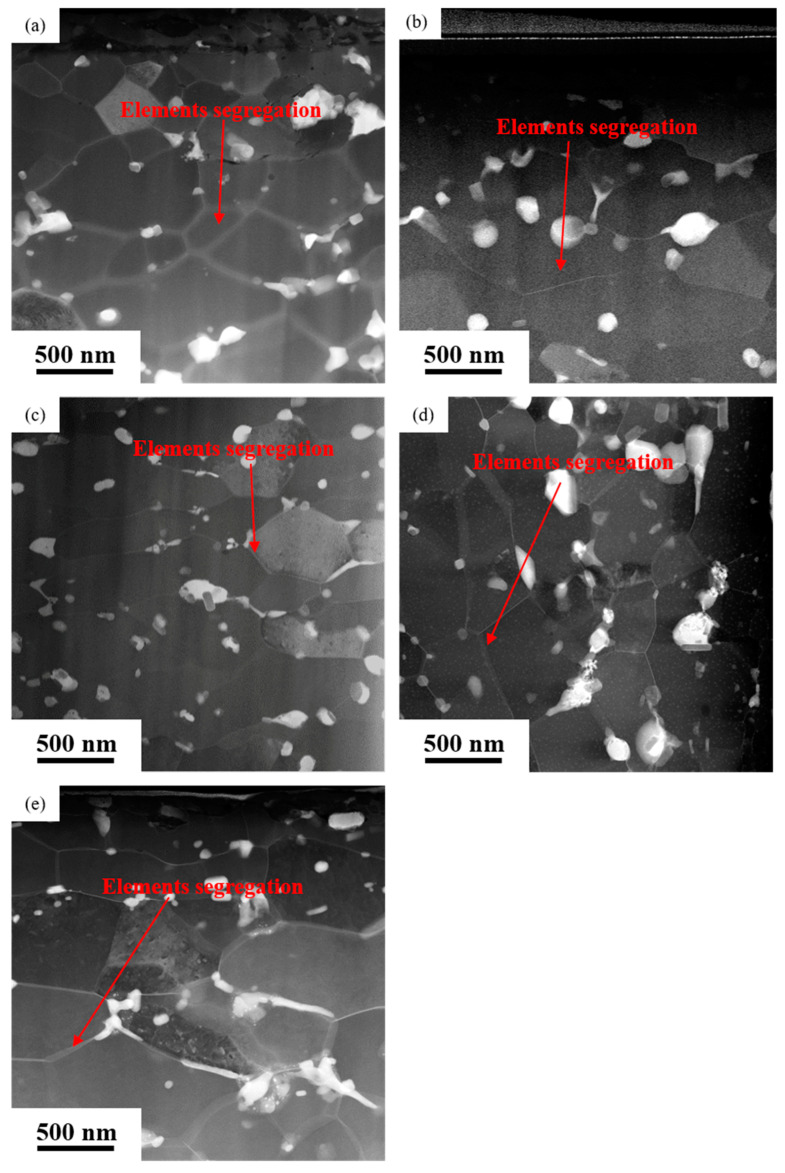
The HAADF micrographs showing the top regions of the near-surface deformed layers on hot rolled AA7050 aluminum alloy under different hot rolling conditions: (**a**) 1# (380 °C, 66.7%); (**b**) 2# (420 °C, 83.3%); (**c**) 3# (420 °C, 66.7%); (**d**) 4# (420 °C, 50%); (**e**) 5# (450 °C, 66.7%).

**Figure 3 materials-16-04632-f003:**
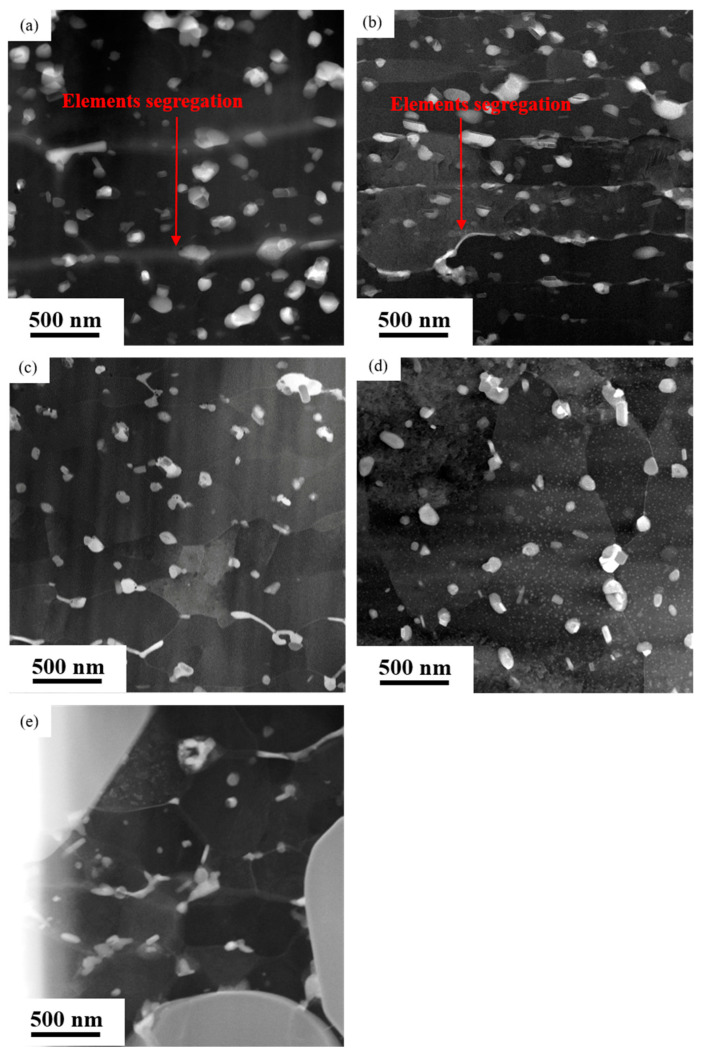
The STEM-HAADF micrographs showing the bottom regions of the near-surface deformed layers on hot rolled AA7050 aluminum alloy under different hot rolling conditions: (**a**) 1# (380 °C, 66.7%); (**b**) 2# (420 °C, 83.3%); (**c**) 3# (420 °C, 66.7%); (**d**) 4# (420 °C, 50%); (**e**) 5# (450 °C, 66.7%).

**Figure 4 materials-16-04632-f004:**
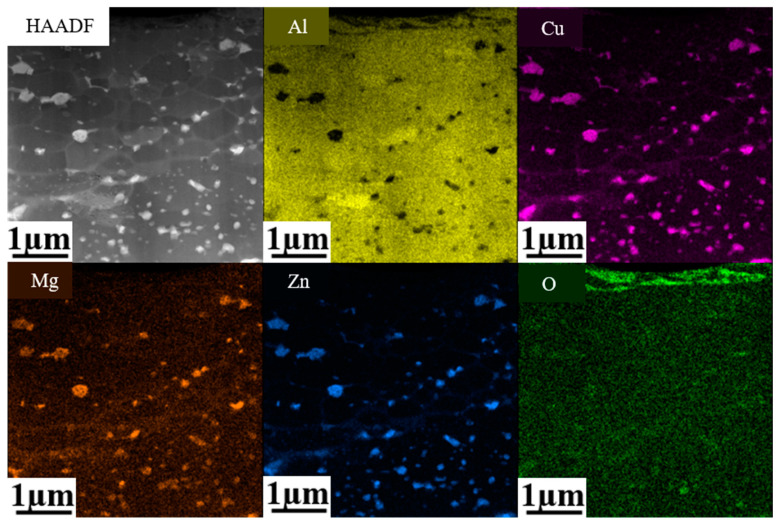
EDS mapping analysis of the near-surface deformed layer of 1# sample (380 °C, 66.7).

**Figure 5 materials-16-04632-f005:**
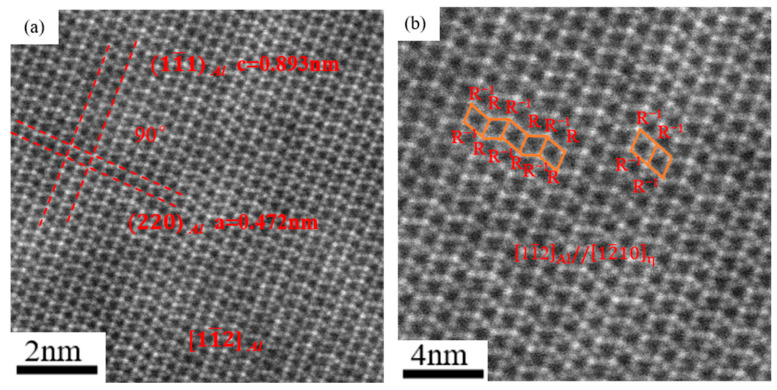
(**a**) The lattice image of the rounded plate precipitate along [1$\overset{–}{1}$2] _Al_ axis; (**b**) the enlarged image in (**a**).

**Figure 6 materials-16-04632-f006:**
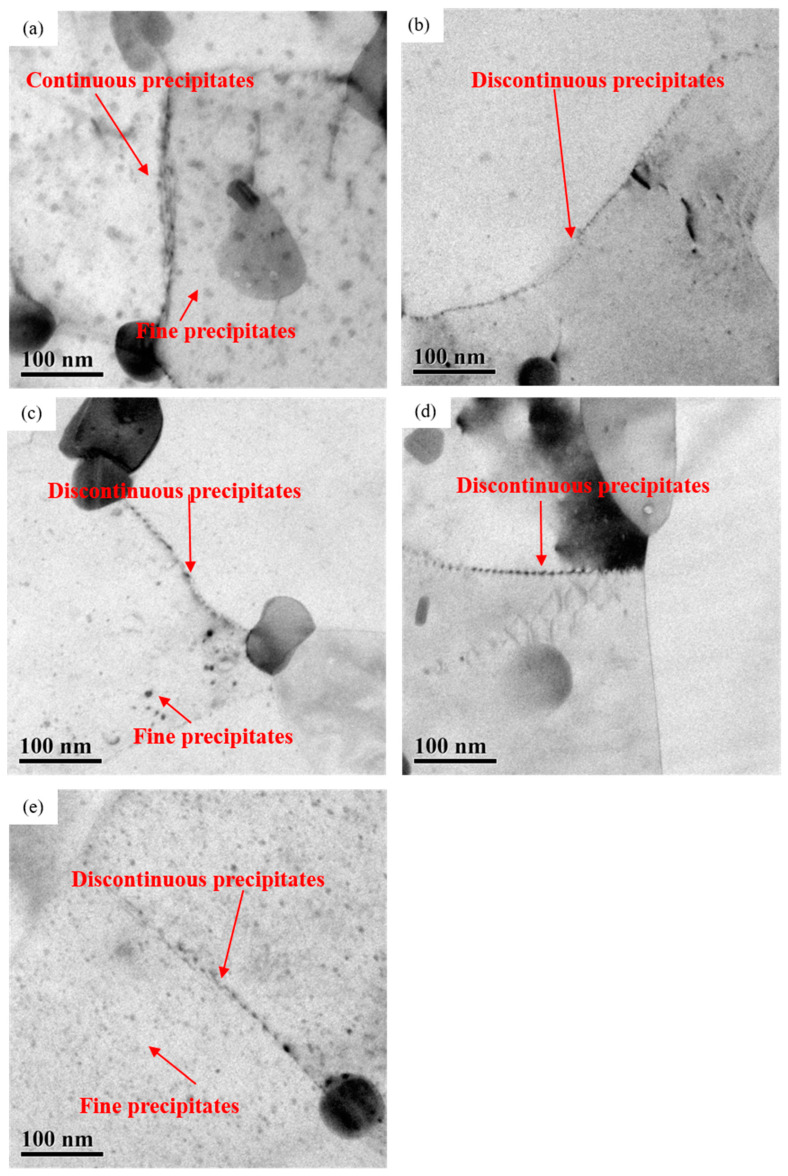
TEM micrographs showing the top regions of the near-surface deformed layers on hot rolled AA7050 aluminum alloy under different hot rolling conditions: (**a**) 1# (380 °C, 66.7%); (**b**) 2# (420 °C, 83.3%); (**c**) 3# (420 °C, 66.7%); (**d**) 4# (420 °C, 50%); (**e**) 5# (450 °C, 66.7%).

**Figure 7 materials-16-04632-f007:**
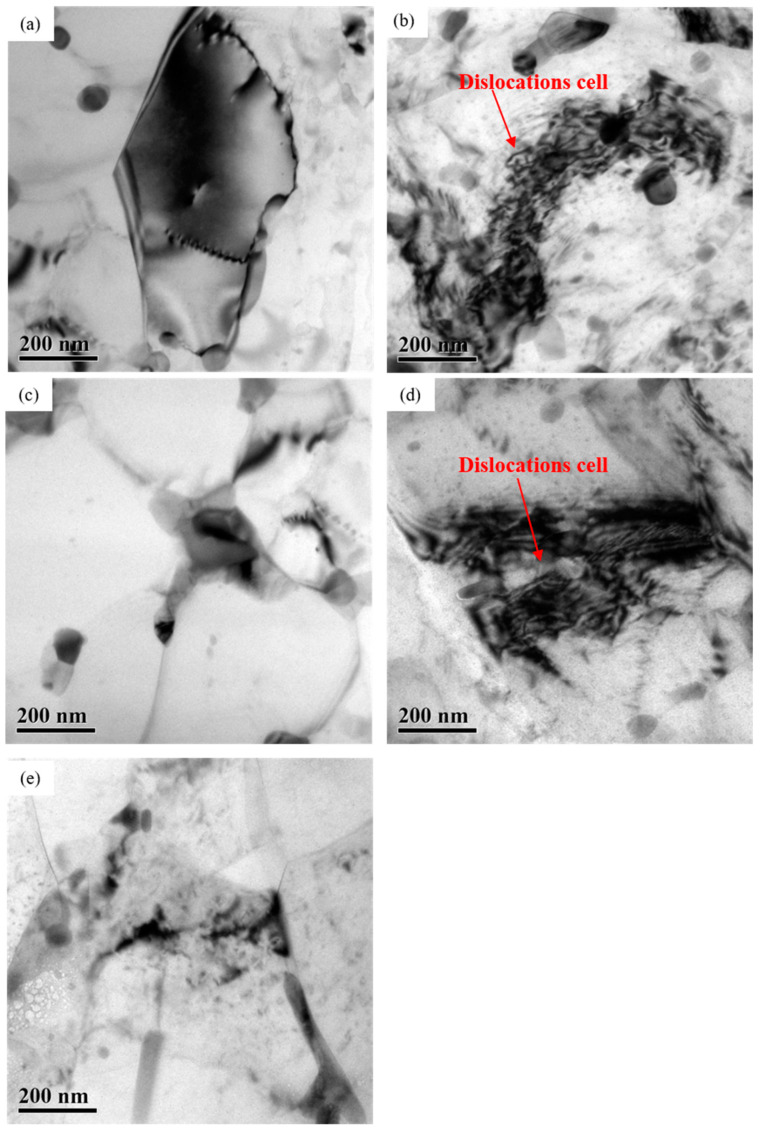
TEM micrographs showing the top regions of the near-surface deformed layers on hot rolled AA7050 aluminum alloy under different hot rolling conditions: (**a**) 1# (380 °C, 66.7%); (**b**) 2# (420 °C, 83.3%); (**c**) 3# (420 °C, 66.7%); (**d**) 4# (420 °C, 50%); (**e**) 5# (450 °C, 66.7%).

**Figure 8 materials-16-04632-f008:**
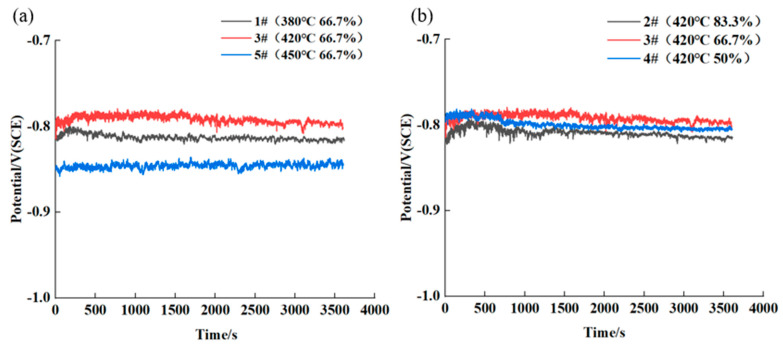
OCP of hot rolled AA7050 aluminum alloy in 3.5 wt.% NaCl solution: (**a**) alloy samples rolled with the same rolling reduction, at different rolling temperature; (**b**) alloy samples rolled at the same hot rolling temperature, with different rolling reduction.

**Figure 9 materials-16-04632-f009:**
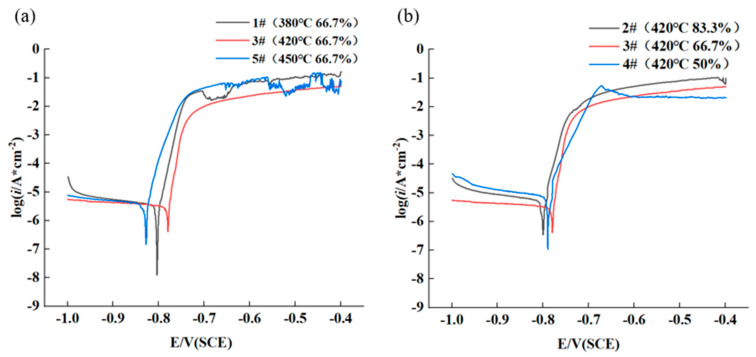
Potentiodynamic polarization curves of hot rolled AA7050 aluminum alloy in 3.5 wt.% NaCl solution: (**a**) the alloy samples rolled with the same rolling reduction, at different hot rolling temperature; (**b**) the alloy samples rolled at the same hot rolling temperature, with different rolling reduction.

**Figure 10 materials-16-04632-f010:**
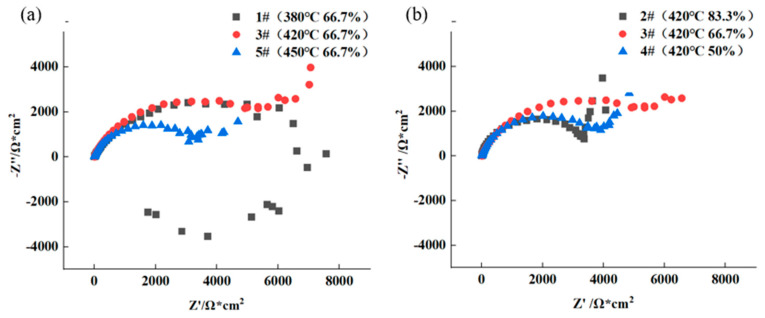
Nyquist diagrams of hot rolled AA7050 aluminum alloy in 3.5 wt.% NaCl solution: (**a**) the alloy samples rolled with the same rolling reduction, at different rolling temperature; (**b**) the alloy samples rolled at the same hot rolling temperature, with different rolling reduction.

**Figure 11 materials-16-04632-f011:**
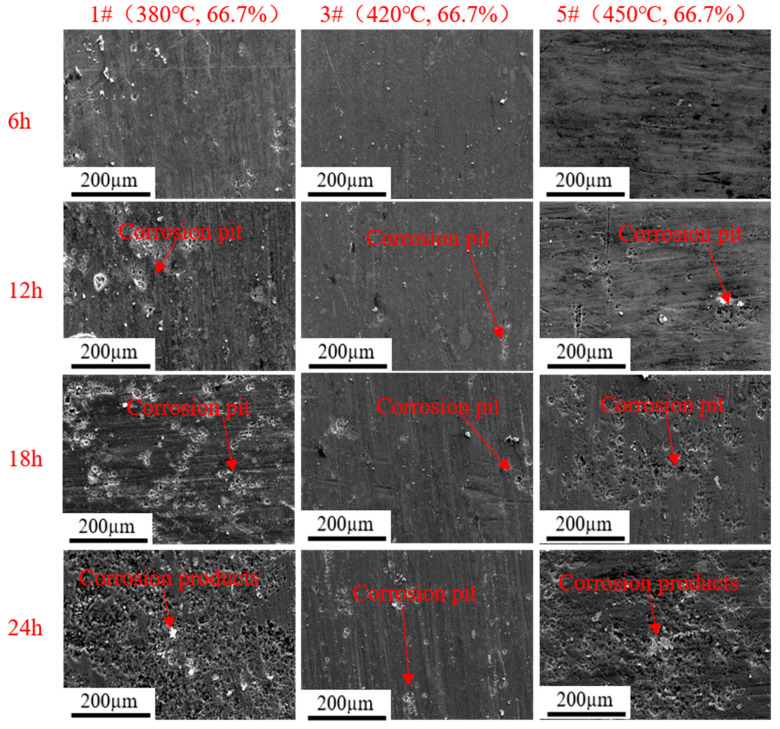
Corrosion morphology of the rolled AA7050 aluminum alloy immersed in 3.5 wt.% NaCl solution for 6, 12, 18, and 24 h (the samples were rolled with the same reduction at different temperatures).

**Figure 12 materials-16-04632-f012:**
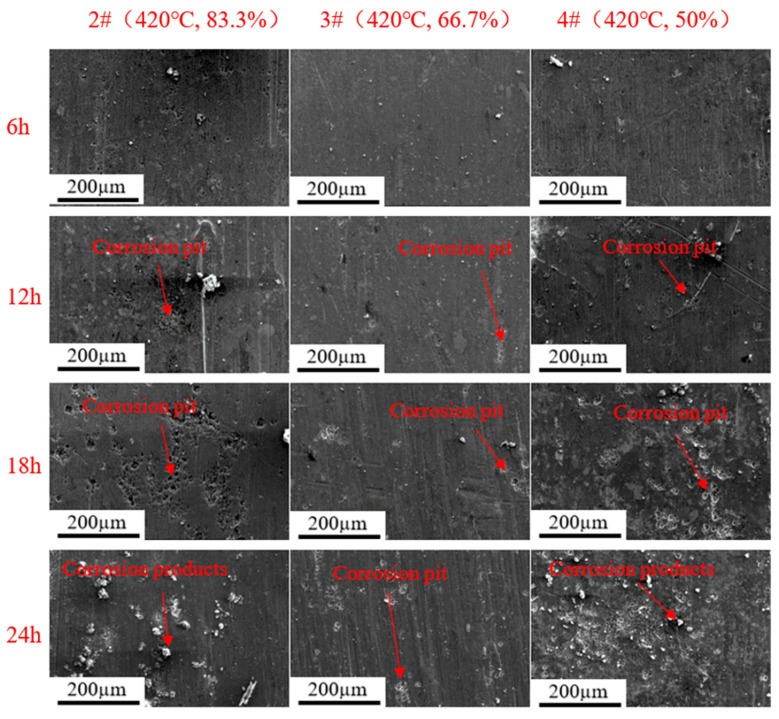
Corrosion morphology of the rolled AA7050 aluminum alloy immersed in 3.5 wt.% NaCl solution for 6, 12, 18, and 24 h (the samples were rolled at the same temperature with different reductions).

**Table 1 materials-16-04632-t001:** Chemical composition of AA7050 aluminum alloy.

	Mg	Zn	Cu	Zr	Cr	Mn	Si	Ti	Fe	Al
Mass fraction/%	2.08	6.07	2.21	0.11	0.02	0.10	0.12	0.04	0.12	Bal.

**Table 2 materials-16-04632-t002:** Hot rolling parameters.

Sample Number	Rolling Temperature/°C	Original Thickness/mm	Thickness After Rolling/mm	Rolling Reduction/%
1#	380	60	20	66.7
2#	420	60	10	83.3
3#	420	60	20	66.7
4#	420	60	30	50.0
5#	450	60	20	66.7

**Table 3 materials-16-04632-t003:** OCP, E_corr_ and I_corr_ of the rolled alloy samples.

Sample	1#	2#	3#	4#	5#
OCP/mV(SCE)	−815.140	−815.081	−802.343	−804.718	−844.650
E_corr_/mV (SCE)	−803.528	−799.255	−778.503	−789.642	−826.569
I_corr_/uA×cm^−2^ (SCE)	1.77	1.58	0.891	1.26	3.16

## Data Availability

Data will be made availability on request.

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
