# Peer review of "Microstructure Evolution of the Near-Surface Deformed Layer and Corrosion Behavior of Hot Rolled AA7050 Aluminum Alloy"

_materials, 2023, doi:10.3390/ma16134632_

Round 1

Reviewer 1 Report

The paper can be accepted, but it requires appropriate revisions to enhance its quality.

Aluminum and aluminum alloys are finding increasing applications in various technical fields. They are used in the aerospace, space, military, automotive, and electronic industries. It would be better to explain the areas of application of aluminum and aluminum alloys. Please refer to the following paper:  https://doi.org/10.18485/aeletters.2018.3.2.2 ,

Expand the introduction with relevant papers from this field.

At the end of the introduction, specify the main contribution of the paper and highlight how it differs from similar papers in this field. What is the reason or significance behind publishing this paper?

Provide the chemical composition of aluminum alloy AA7050.

Provide a better explanation for figures 1, 2, and 3.

How many different samples were used to obtain the diagrams in figures 9 and 10? Explain the obtained results.

Enlarge figures 11 and 12.

Based on an expanded analysis and discussion, expand the concluding remarks.

Author Response

Dear Reviewer:

Thank you for your letter and the reviewers’ comments concerning our manuscript (ID: materials-2444688) entitled “Microstructure evolution of the near-surface deformed layer and corrosion behavior of hot rolled AA7050 aluminum alloy”. We have made changes of our manuscript, based on the comments. The main corrections are in the manuscript and the response to the reviewers’ comments are detailed in the attachment below (the replies are highlighted in blue).

Reviewer 2 Report

The article is devoted to the analysis of microstructure evolution of a deformed layer in the AA7050 aluminum alloy. The subject of this study is relevant, and the results are common. The introduction substantiates the purpose of the work quite well, but the authors refer only to the work of their Chinese colleagues. I believe that the literature review can be expanded. In particular, one can consider the work of an international group of scientists who study the aluminum alloys.

Despite the good level of this research, I have a few comments that should be corrected at the beginning of the publication process. The main ones are possible:

Materials and methods section

1) It is not stated how many passes were used to obtain the final thickness of samples. One pass or several? If more than one, was the samples heated between passes?

2) Please provide more information of rolling parameters and equipment. For example, rolls speed, roll diameter, etc.

3) It is necessary to add information on the preparation of samples for microstructural studies. Also indicate in which areas the microstructure analysis was carried out.

 4) Please explain why exactly these temperature and deformation parameters were chosen for hot rolling of the AA7050 alloy?

Results Section

5) Please add the size chart in FIG. 5b.

Discussion Section

6) Lines 304–310.

In this case, if the temperature of the rolls was approximately equal to the ambient temperature, and taking into account the cooling of the surface of the samples in air, it cannot be argued that the temperature decreases from the surface to the center. On the contrary, the surface temperature should be lower than the temperature of the central layers.

 7) Section 4.2 is considered to have a more detailed discussion of the relationship between microstructure formation and corrosion resistance. Why is Sample #3 exactly the best corrosion performance?

The English text is written quite well. However, there are a few typos in the text.

Author Response

(The authors gave the same response as above.)
